# Proximity labeling proteomics reveals critical regulators for inner nuclear membrane protein degradation in plants

Aobo Huang[1], Yu Tang[2], Xuetao Shi[1], Min Jia[2], Jinheng Zhu[1], Xiaohan Yan[1], Huiqin Chen[1] & Yangnan Gu [2,3✉]

The inner nuclear membrane (INM) selectively accumulates proteins that are essential for nuclear functions; however, overaccumulation of INM proteins results in a range of rare genetic disorders. So far, little is known about how defective, mislocalized, or abnormally accumulated membrane proteins are actively removed from the INM, especially in plants and animals. Here, via analysis of a proximity-labeling proteomic profile of INM-associated proteins in *Arabidopsis*, we identify critical components for an INM protein degradation pathway. We show that this pathway relies on the CDC48 complex for INM protein extraction and 26S proteasome for subsequent protein degradation. Moreover, we show that CDC48 at the INM may be regulated by a subgroup of PUX proteins, which determine the substrate specificity or affect the ATPase activity of CDC48. These PUX proteins specifically associate with the nucleoskeleton underneath the INM and physically interact with CDC48 proteins to negatively regulate INM protein degradation in plants.

[1] Tsinghua-Peking Joint Center for Life Sciences, Center for Plant Biology, School of Life Sciences, Tsinghua University, Beijing, China. [2] Department of Plant and Microbial Biology, University of California, Berkeley, CA, USA. [3] Innovative Genomics Institute, University of California, Berkeley, CA, USA. ✉email: guyangnan@berkeley.edu

The nuclear envelope (NE) is composed of two concentric lipid bilayers, the outer nuclear membrane (ONM) that is continuous with the endoplasmic reticulum (ER), and the inner nuclear membrane (INM) that faces the nucleoplasm. The ONM is jointed with the INM at the nuclear pore, where the nuclear pore complex (NPC) resides. As a distinct membrane territory, the INM hosts a unique set of membrane proteins that are required for essential nuclear functions, such as genome organization, transcriptional control, mechanosensation, and signal transduction[1]. However, abnormal accumulation of INM proteins has been associated with altered nuclear morphology and a range of rare genetic diseases[2,3]. For example, overaccumulation of the evolutionarily conserved INM protein SUN1 (Sad1 and UNC84 Domain Containing 1) is pathogenic in humans and is linked to muscular dystrophy and premature aging syndrome[4]. Despite the obvious importance in maintaining protein homeostasis and integrity at the INM, we know very little about the mechanism that removes abnormally accumulated or defective membrane components from the INM, especially in plants and animals.

Previous studies in yeasts have uncovered an INM-associated degradation (INMAD) pathway that mediates destruction of integral INM proteins[5,6]. Nevertheless, key components in yeast INMAD pathway are missing in plants and animals. For example, the initiation of yeast INMAD relies on membrane-bound E3 ubiquitin ligases Asi1 and Asi3, which localize to the INM and ubiquitinate misfolded or mislocalized INM proteins for proteasomal degradation[5,6]. However, Asi genes do not have homologs in plants and animals. Doa10 is another E3 ligase found in ER and INM and participates in both ER-associated degradation (ERAD) and INMAD in yeasts[7]. In addition, the Anaphase-promoting complex (APC/C) mediates the degradation of INM protein Mps3 in yeasts[8]. Nonetheless, neither Doa10 nor APC/C homolog has been linked to INMAD in plants or animals. Recently, it was reported that the human INM protein Emerin is selectively cleared under ER stress through vesicular transport to lysosomes, but this process depends on Emerin's LEM domain and is not observed for other INM proteins[9]. So far, we still miss key evidence to support the existence of a ubiquitin/proteasome-dependent INMAD pathway outside of yeasts.

Proximity-labeling-based proteomic approaches have been recently applied to plant research and have been shown to be invaluable in profiling functional components of various protein complexes[10–14]. By combining the power of BioID2-based proximity labeling (PL)[15], label-free quantitative mass spectrometry (LFQMS), and ratiometric analysis (RA), we profiled the NE-associated proteome in *Arabidopsis* using known NE proteins as bait[16]. Here, we report a group of periphery NE proteins that are explicitly associated with the INM and involved in ubiquitin-mediated proteolysis. They include CDC48 proteins, its cofactors UFD1 and NPL4, and a specific subgroup of plant ubiquitin regulatory X (UBX) domain-containing proteins (PUXs). CDC48, also known as p97 in mammals, is a conserved AAA-ATPase molecular chaperon that can mediate the extraction of integral proteins from the membrane and recruit 26S proteasome for subsequent protein degradation[17,18]. PUX proteins define a plant protein family that possesses a conserved UBX domain, which mediates direct interactions with CDC48 proteins[19]. Some PUXs also contain a ubiquitin-associated (UBA) domain, which allows them to bind ubiquitinated protein substrates and act as selective adapters between CDC48 and membrane substrates[20–22]. However, PUX without the UBA domain has been shown to directly interfere with the CDC48 activity[23–25].

We used the conserved INM protein SUN1 as a model to dissect the functional connection between the INM-CDC48 association and INM protein degradation in plants. We showed

that SUN1 undergoes constitutive degradation in a proteasome-dependent and autophagy-independent manner in *Arabidopsis*, and this process engages the CDC48 complex. Moreover, we showed that the CDC48 activity at the INM may be directly regulated by a specific subgroup of PUX proteins, including PUX3, PUX4, and PUX5. These PUX proteins have evolved a membrane preference for the INM through association with the nucleoskeleton and physically interact with CDC48 proteins to negatively regulate INM protein degradation in plants.

## Results

### Identification of proteins specifically associated with the INM.
Previously we fused a promiscuous biotin ligase (BioID2) to AtSUN1, a conserved and one of the best characterized integral INM resident protein, and performed PL-LFQMS experiment using a *35S: HA-BioID2-SUN1* transgenic line to profile INM-associated proteins[16]. Here, we performed additional validation, including the relative protein expression level, NE localization, and inducible biotinylation of HA-BioID2-SUN1 in the transgenic line to further support the specificity and efficiency of our previous PL-LFQMS profiling (Supplementary Fig. 1 and Supplementary Note 1). To identify candidates that are specifically associated with SUN1 at the INM and exclude those that associate with both the INM and the ONM, we reanalyzed the SUN1 PL-LFQMS data using the ONM-anchored protein WIT1[26] as a control (Fig. 1a). We generated transgenic plants expressing BioID2-tagged WIT1 and performed PL-LFQMS. The resulting MS data together with the available MS data from BioID2-SUN1 and plants expressing YFP-BioID2 without biotin treatment (Mock)[16] were used for a three-dimensional ratiometric analysis. The peptide intensities were normalized across SUN1, WIT1, and Mock samples and were used to perform pairwise comparisons (Fig. 1b). Using *p*-value < 0.05 and fold-change > 2 as cutoffs for both controls, we obtained 15 SUN1-specific preys (Fig. 1c and Supplementary Fig. 2a). Among them, five are components of the plant nucleoskeleton, including CRWN1, CRWN2, CRWN3, CRWN4, and KAKU4, consistent with the well-established interaction between SUN1 and the nucleoskeleton at the INM[27,28]. On the other hand, we identified a total of four WIT1-specific preys, including WIP1, WIP3, RanGAP1, and RanGAP2 (Fig. 1c and Supplementary Fig. 2b), supporting previous reports that WIT1 forms complexes with WIP proteins and anchors RanGAPs to the ONM[26,29,30]. These analyses reinforce the high specificity of PL-LFQMS-RA in identifying local proteome in plants[16]. An improved biotin ligase enzyme with higher labeling efficiency termed TurboID has been recently developed[13,14,31]. Using TurboID in PL-LFQMS-RA may further improve its efficiency and result in a more comprehensive profiling.

### The CDC48 complex associates with integral INM proteins.
Intriguingly, among the SUN1-specific preys, four proteins are potentially associated with the molecular chaperone CDC48, including two ubiquitin fusion degradation (UFD) proteins (UFD1B and UFD1C) and two PUX proteins (PUX4 and PUX5) (Fig. 1c). In yeasts and mammals, UFD proteins are part of a CDC48-UFD-NPL4 heterotrimeric complex, which mediates the membrane extraction of ubiquitinated proteins and recruits 26S proteasome for subsequent protein degradation[17,18]. PUX proteins also can directly interact with CDC48 and function to regulate CDC48 activity or to bridge CDC48 with ubiquitinated membrane protein substrates[20,23–25,32,33].

To verify the connection of the identified UFD and PUX proteins with CDC48, we performed yeast-two-hybrid experiments. We found that PUX4, PUX5, UFD1B, and UFD1C interacted with three *Arabidopsis* CDC48 paralogs, except that the

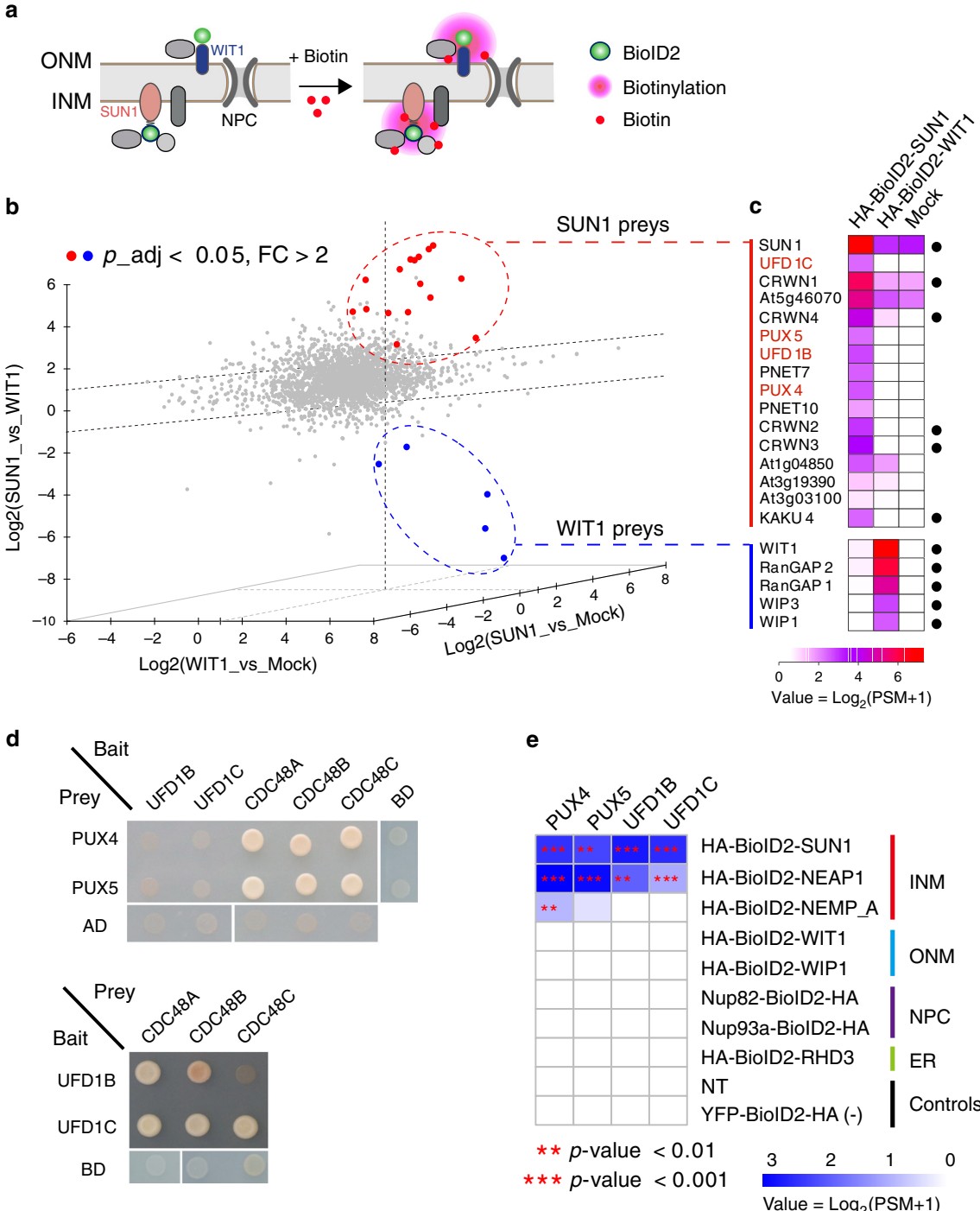

**Fig. 1 The CDC48-dependent proteolysis machinery associates with integral INM proteins. a** Identification of proteins that are specifically associated with SUN1 at the INM using proximity labeling. The ONM protein WIT1 serves as a control to exclude proteins associated with both the INM and the ONM. **b** Pairwise ratiometric analyses of PL-LFQMS data using *35S: HA-BioID2-SUN1* (+biotin), *35S: HA-BioID2-WIT1* (+biotin), and *35S: YFP-BioID2-HA* (−biotin) samples. Three biological replicates for each sample were included in the analyses. Axes represent the log values of peptide intensity ratio (FC, fold-change) between two samples. Significantly enriched proteins in one sample were defined by cutoffs FC > 2 and adjusted *p*-value < 0.05 (linear model *F*-test) relative to both controls. Proteins significantly enriched in HA-BioID2-SUN1 and HA-BioID2-WIT1 samples are labeled in red and blue dots, respectively. **c** Heatmap showing transformed and averaged peptide spectrum match (PSM) values of significantly enriched proteins in HA-BioID2-SUN1 and HA-BioID2-WIT1 samples. Known SUN1 and WIT1 interacting proteins are marked by black dots on the right. **d** Yeast-two-hybrid analyses among PUX4/5, UFD1B/1C, and CDC48A/B/C. Diploid yeasts after mating were grown on QDO medium (SD-Leu/-Trp/-His/-Ade) for 3 days. Similar results have been obtained twice. **e** Transformed and averaged PSM data of PUX4, PUX5, UFD1B, and UFD1C from PL-LFQMS experiments using BioID2-tagged bait proteins from the inner nuclear membrane (INM), the outer nuclear membrane (ONM), the nuclear pore complex (NPC), and the endoplasmic reticulum (ER). Three biological replicates for each sample were included for analysis. Biotin-treated WT non-transgenic plants (NT) and YFP-BioID2-HA plants without biotin treatment were served as controls. Asterirks (** and ***) represent significant enrichment with *p*-value < 0.01 and 0.001 (linear model *F*-test), respectively, compared to both controls. MS data are available through PRIDE (Identifiers PXD015919 and PXD015920) and see Supplementary Data 2 for the dataset list.

interaction between UFD1B and CDC48C might be weak (Fig. 1d). This result supports the idea that PUX4, PUX5, UFD1B, and UFD1C may directly participate in the CDC48-dependent proteolysis pathway in plants.

Notably, when we performed reanalyses on a series of previously published PL-LFQMS profiling datasets using NE proteins as baits[16], we found that UFD1B, UFD1C, PUX4, and PUX5 were also probed by another INM protein NEAP1 (Fig. 1e and Supplementary Fig. 2c). In contrast, none of these UFD and PUX proteins were detected by other baits, including the ONM protein WIP1 and WIT1, the NPC component Nup93a and Nup82, and the ER membrane protein RHD3. In addition, PL-LFQMS profiling was performed using a third INM bait protein NEMP_A, an Arabidopsis ortholog of the Xenopus INM protein Nemp1, and PUX4 and PUX5 were also probed (Fig. 1e and Supplementary Fig. 2c). These data are consistent with the idea that there is a specific association of INM proteins with the identified CDC48-dependent membrane protein degradation machinery.

**INM protein SUN1 undergoes proteasome-dependent degradation.** The degradation of INM proteins and associated regulatory mechanisms have not been defined in plants or metazoans. To gain functional insight into the association of INM proteins with UFD and PUX proteins and its potential connection with the INMAD, we first assayed whether INM protein homeostasis involves proteasomal degradation in plants. We found that treatment of 35S: HA-BioID2-SUN1 seedlings with the proteasome inhibitor MG132 significantly increased the steady-state accumulation of SUN1 (Fig. 2a). Moreover, MG132 treatment also induced detectable accumulation of polyubiquitinated SUN1 (Fig. 2b), suggesting that SUN1 may undergo ubiquitin/proteasome-dependent degradation. Next, we set up an in vivo protein degradation assay. We found that when the 35S: HA-BioID2-SUN1 seedlings were treated with cycloheximide (CHX), which blocks the de novo protein synthesis, the level of SUN1 decayed with time, suggesting that the degradation is constitutive (Fig. 2c). Adding MG132 to CHX-treated plants significantly inhibited SUN1 degradation over time (Fig. 2d), whereas adding concanamycin A to block autophagy did not affect SUN1 degradation (Fig. 2e). These results suggest that at least some INM proteins are degraded through a proteasome-dependent but autophagy-independent degradation pathway in plants.

**The entire CDC48 complex was captured by SUN1.** Knowing that the proteasome participates in SUN1 degradation, we performed a second round of PL-LFQMS profiling using MG132-treated HA-BioID2-SUN1 plants to trap the degradation machinery. Remarkably, MG132 treatment enabled HA-BioID2-SUN1 to capture all core components of the CDC48-UFD-NPL4 trimeric complex, including CDC48B, CDC48C, UFD1B and UFD1C, and NLP4A (Fig. 2f and Supplementary Fig. 3a). This result strongly supports the involvement of the CDC48 complex in the proteasome-dependent INM protein degradation.

**PUX3/4/5 selectively associate with the INM.** Beside the entire CDC48 complex, MG132 treatment allowed HA-BioID2-SUN1 to capture a third PUX protein (PUX3) in addition to PUX4 and PUX5 (Fig. 2f and Supplementary Fig. 3a). Intriguingly, PUX3, PUX4, and PUX5 are closely related and form a subclade within the 16 PUXs encoded by Arabidopsis (Fig. 2g and Supplementary Fig. 3b). Emerging evidence suggests that the 16 AtPUX proteins play an important role in determining the substrate specificity and ATPase activity of CDC48 and are functionally diversified by associating with different membrane compartments. For example,

PUX10 explicitly localizes to the lipid droplet (LD) membrane via a unique hydrophobic polypeptide sequence and recruits CDC48 for LD protein degradation[21,22]. PUX7, PUX8, PUX9, and PUX13 associate with the autophagosome by interacting with ATG8 through their ubiquitin-interacting motif (UIM)-like sequence and recruit inactive CDC48 molecules for degradation[34]. In our PL-LFQMS experiments, INM baits only probed PUX members in the PUX3/4/5 subclade (Fig. 3a and Supplementary Fig. 4a). Because these PUX genes are not expressed at a higher level than other PUXs (Supplementary Fig. 4b), identification of PUX3/4/5 suggests their specific function at the INM. In line with this observation, the PUX3/4/5 subgroup is homologous to the yeast UBX protein UBX1 (Supplementary Fig. 3b), which is involved in the degradation of yeast INM protein Asi1[35].

To confirm the specific association of PUX3/4/5 with the INM, we transiently coexpressed SUN1 with PUX4, PUX5, and PUX7, respectively. Co-immunoprecipitation (co-IP) assay showed that SUN1 was specifically co-purified with PUX4 and PUX5, but not PUX7 (Fig. 3b). We also examined the association of PUX4 with proteins from different membrane compartments. We transiently coexpressed PUX4 with four membrane proteins, including SUN1 at the INM, WIT1 at the ONM, BRI1 at the plasma membrane, and ARA6 at the late endosome. Again, the co-IP result demonstrated that PUX4 specifically pulled down SUN1 but not the other membrane proteins tested (Fig. 3c). However, when we used bimolecular fluorescence complementation (BiFC) assay to verify the interaction between SUN1 and PUX4, only very weak signal was detected (Supplementary Fig. 4c), suggesting that PUX3/4/5 may not interact with INM proteins constitutively but in a transient or indirect manner. Nevertheless, these data further support the specific association of PUX3, PUX4, and PUX5 with the INM.

**PUX3/4/5 interact with the nucleoskeleton.** To investigate how PUX3/4/5 are selectively associated with the INM, we generated 35S: PUX5-BioID2 transgenic plants and performed PL-LFQMS profiling. Among the top candidates, we identified both PUX3 and PUX4 but no other PUXs, suggesting that PUX3, PUX4, and PUX5 may function together at the INM (Fig. 3d). We also identified components of the CDC48 complex, including CDC48A, CDC48B, UFD1B, UFD1C, and NPL4A, reinforcing the direct interaction between PUX3/4/5 and the CDC48 complex (Fig. 1d).

Importantly, CRWN4, a component of the nucleoskeleton, was identified as a specific PUX5 interactor with high confidence (Fig. 3d). To confirm the interaction of PUX3, PUX4, and PUX5 with the nucleoskeleton, we performed BiFC assay between PUX3/4/5 and another nucleoskeleton component KAKU4. Different from the BiFC signal between PUX4 and SUN1, which was only weakly detected, robust complemented fluorescence was observed at the nuclear periphery between PUX3/4/5 and KAKU4 (Fig. 3e), suggesting a stable association of PUX3, PUX4, and PUX5 with the nucleoskeleton. This data implicates a mechanism for the INM targeting of PUX3, PUX4, and PUX5 and explains their proximity with multiple INM proteins.

**PUX3/4/5 negatively regulate SUN1 degradation.** To investigate the functional importance of PUX3, PUX4, and PUX5 in INM protein degradation, we took advantage of the CRISPR/Cas9 gene-editing tool and generated a series of pux mutants in an isogenic 35S: HA-BioID2-SUN1 background, including a pux3 single mutant, a pux3 pux4 and a pux3 pux5 double mutant, and a pux3 pux4 pux5 triple mutant. The Cas9-generated mutations were all nonsense and occurred near the beginning of target genes, which led to frameshifts and premature stop codons, likely

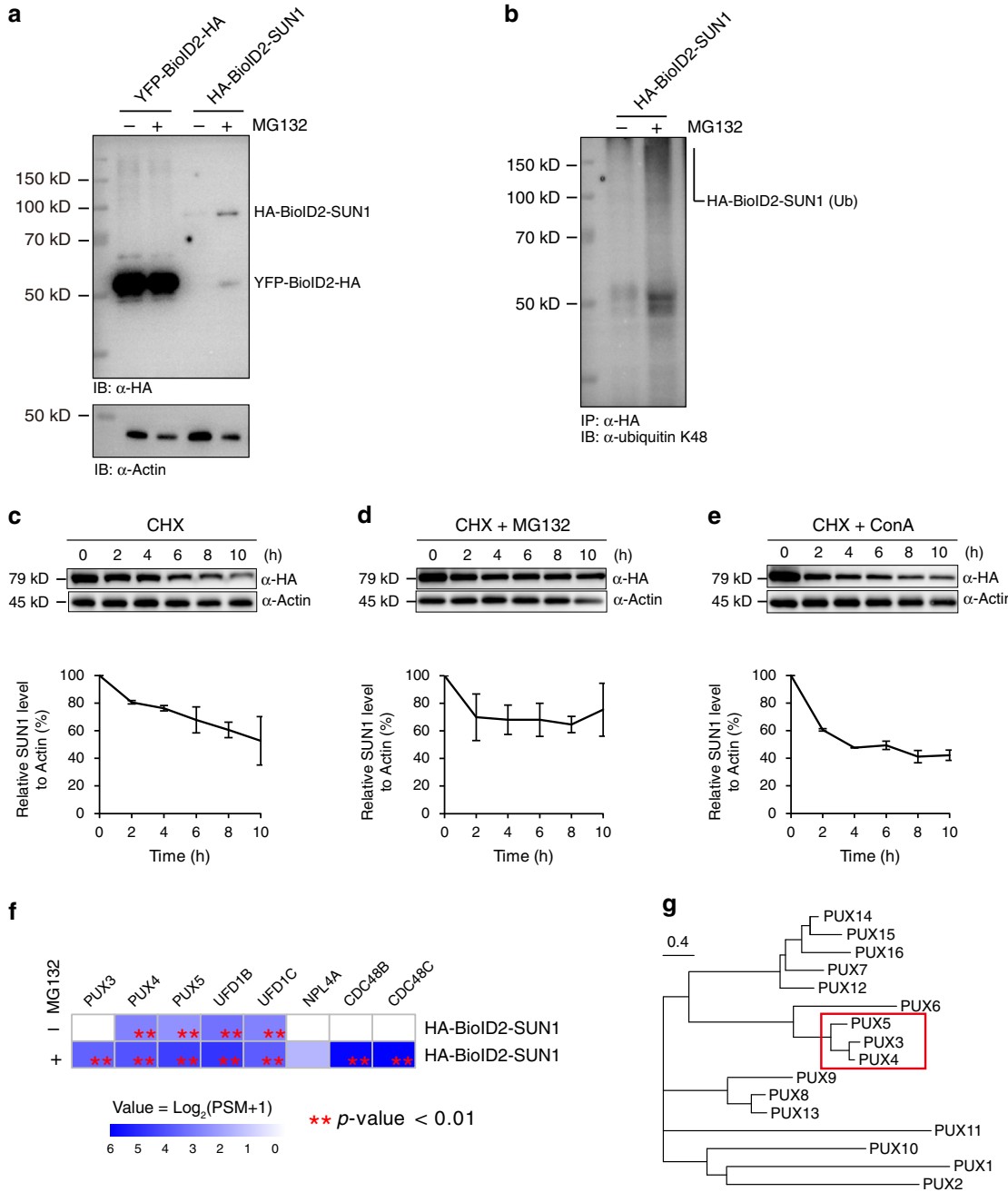

**Fig. 2 SUN1 is subject to proteasome-dependent but autophagy-independent degradation. a** Steady-state accumulation of HA-BioID2-SUN1. Ten-day-old *35S: HA-BioID2-SUN1* seedlings were treated with or without 50 μM MG132 for 8 h before total protein was extracted and immunoblotted with anti-HA and anti-actin antibodies. The *35S: YFP-BioID2-HA* transgenic plants were used as the control. **b** Total protein from the mock-treated or MG132-treated *35S: HA-BioID2-SUN1* seedlings was first immunoprecipitated with anti-HA antibody-coated agarose beads and then immunoblotted with anti-ubiquitin antibody (K48 linkage). **c** In vivo protein degradation assay for HA-BioID2-SUN1. Ten-day-old seedlings expressing HA-BioID2-SUN1 were treated with 100 μM CHX for the indicated times before total protein was extracted and immunoblotted with anti-HA and anti-actin antibodies. **d** In vivo HA-BioID2-SUN1 degradation was blocked when 50 μM MG132 was used to treat plants together with CHX. **e** In vivo HA-BioID2-SUN1 degradation was not affected when 1 μM ConA was used to treat plants together with CHX. **c–e** The lower panels are densitometric analyses of HA-BioID2-SUN1 protein levels using actin as the control. Error bars represent standard deviations (SD) from two independent biological replicates. **f** Transformed and averaged PSM values of the CDC48 complex components and PUX3/4/5 from PL-LFQMS profiling using *35S: HA-BioID2-SUN1* plants with or without MG132 treatment. Two biological replicates were used. Asteriks (**) represents significant enrichment with *p*-value < 0.01 (linear model *F*-test) compared to controls. **g** Phylogenetic analysis of 16 PUX members in *Arabidopsis*. The neighbor-joining tree was generated using amino acid sequences. Bar corresponds to substitution per site.

resulting in loss-of-function mutations (Fig. 4a and Supplementary Fig. 5a). Next, we treated *HA-BioID2-SUN1* seedlings with CHX in WT and *pux* mutant background in parallel to compare the turnover rate of SUN1 protein. Surprisingly, we found that

the SUN1 turnover rate is faster in the *pux3* single mutant than that in WT, although SUN1 steady-state accumulation is unchanged. This effect was further enhanced in the *pux3 pux4* and *pux3 pux5* double mutants (Supplementary Fig. 5b). The

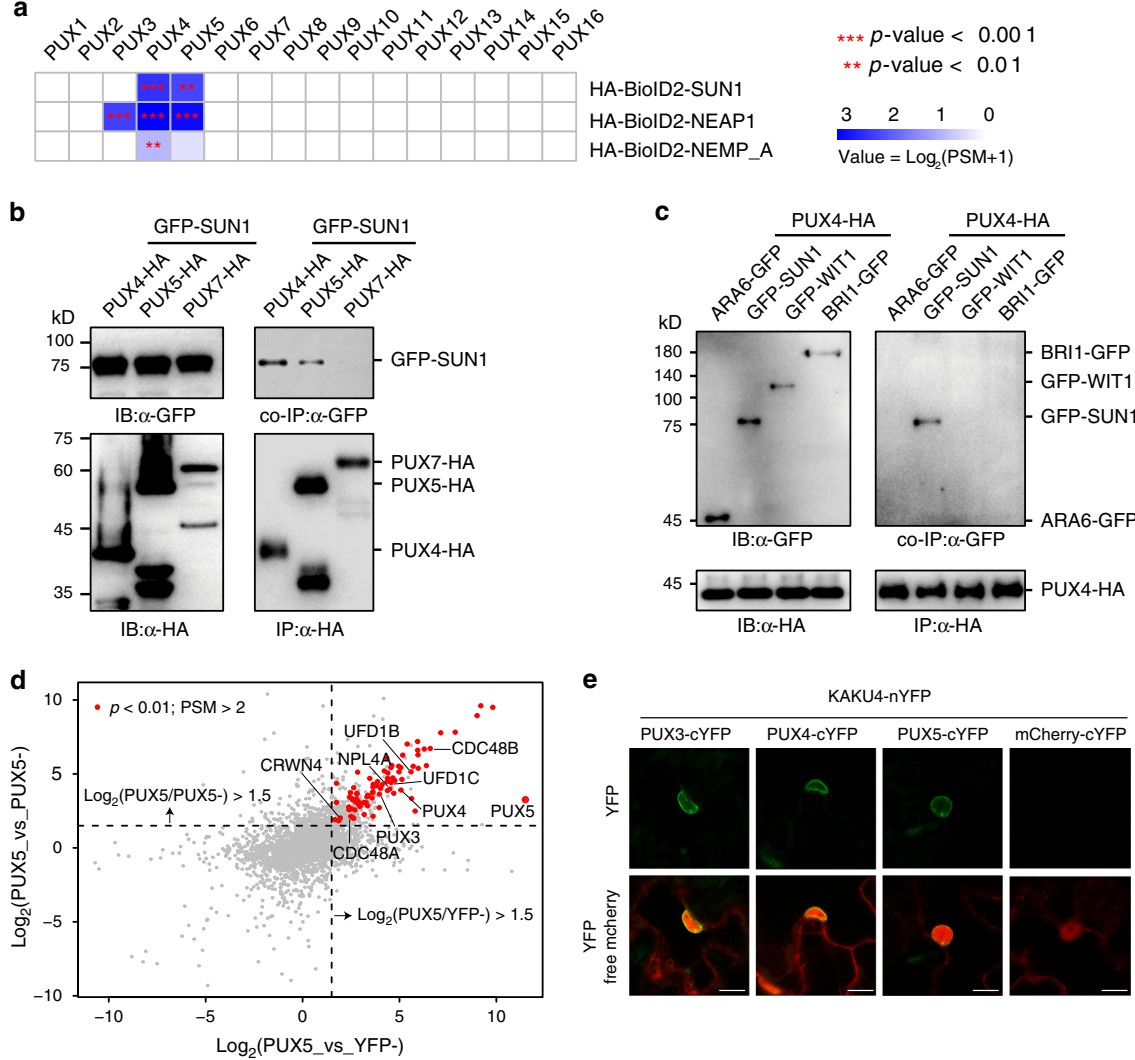

**Fig. 3 PUX3/4/5 selectively associate with the INM via interaction with the nucleoskeleton. a** Transformed and averaged PSM data of all AtPUX proteins from PL-LFQMS profiling using *Arabidopsis* plants expressing BioID2-tagged INM protein SUN1, NEAP1, and NEMP_A. Three biological replicates were used. Asterisks (**, ***) represent significant enrichment with *p*-value < 0.01 and 0.001 (linear model *F*-test) compared to controls. **b** SUN1 specifically associates with PUX4 and PUX5 but not PUX7. GFP-SUN1 was transiently coexpressed with PUX4-HA, PUX5-HA, or PUX7-HA in *N. benthamiana*. Protein extract was immunoprecipitated with anti-HA antibody-conjugated agarose beads before immunoblotting with anti-HA and anti-GFP antibodies. **c** PUX4 associates with SUN1 but not with ARA6, WIT1, and BRI1. PUX4-HA was transiently coexpressed with GFP-SUN1, GFP-WIT1, ARA6-GFP, or BRI1-GFP in *N. benthamiana*. Protein extract was immunoprecipitated with anti-HA antibody-conjugated agarose beads before immunoblotting with anti-HA and anti-GFP antibodies. **d** Scatter plot showing significantly enriched proteins (red dots) identified by PL-LFQMS profiling using PUX5 as the bait. PUX5-BioID2-HA without biotin treatment (PUX5−) and YFP-BioID2-HA without biotin treatment (YFP−) were used as controls. Three biological replicates were used. Candidates with PSM > 2 were further filtered using cutoffs log$_2$FC > 1.5 and *p*-value < 0.01 (linear model *F*-test) compared to both controls. **e** BiFC assays showing interactions between PUX3/4/5 and KAKU4 at the nuclear periphery. BiFC YFP constructs together with free mCherry were transiently coexpressed in *N. benthamiana*, and epidermal cells were imaged. Bars = 10 μm.

accelerated turnover rate was most dramatic in the *pux3 pux4 pux5* triple mutant, which showed a nearly 70% reduction in SUN1 protein level within 4 h of CHX treatment (Fig. 4b, c). Nevertheless, MG132 treatment greatly compromised the SUN1 degradation in the *pux3 pux4 pux5* triple mutant (Supplementary Fig. 5c), similar to what was observed in WT plants (Fig. 2d), suggesting that the regulation by PUX3/4/5 is upstream of the proteasome.

Consistent with the enhanced degradation of SUN1 in the absence of PUX3, PUX4, and PUX5, when PL-LFQMS was performed using *HA-BioID2-SUN1* plants in WT and *pux3 pux4 pux5* mutant background, we observed a significant decrease in the biotinylation level of the nucleoskeleton, including CRWN1,

CRWN2, and CRWN3, in the *pux3 pux4 pux5* mutant compared to WT (Fig. 4d). Because that the steady-state SUN1 accumulation was not obviously affected in the triple mutant, the reduced capacity of SUN1 in labeling its proximal proteins may attribute to its reduced stability and half-life in the INM in the absence of PUX3, PUX4, and PUX5. Together, these data demonstrated that PUX3, PUX4, and PUX5 are redundantly required for the protection of INM protein from proteasomal degradation.

## Discussion

The membrane-associated protein degradation mechanisms are essential for the maintenance of integrity and identity of

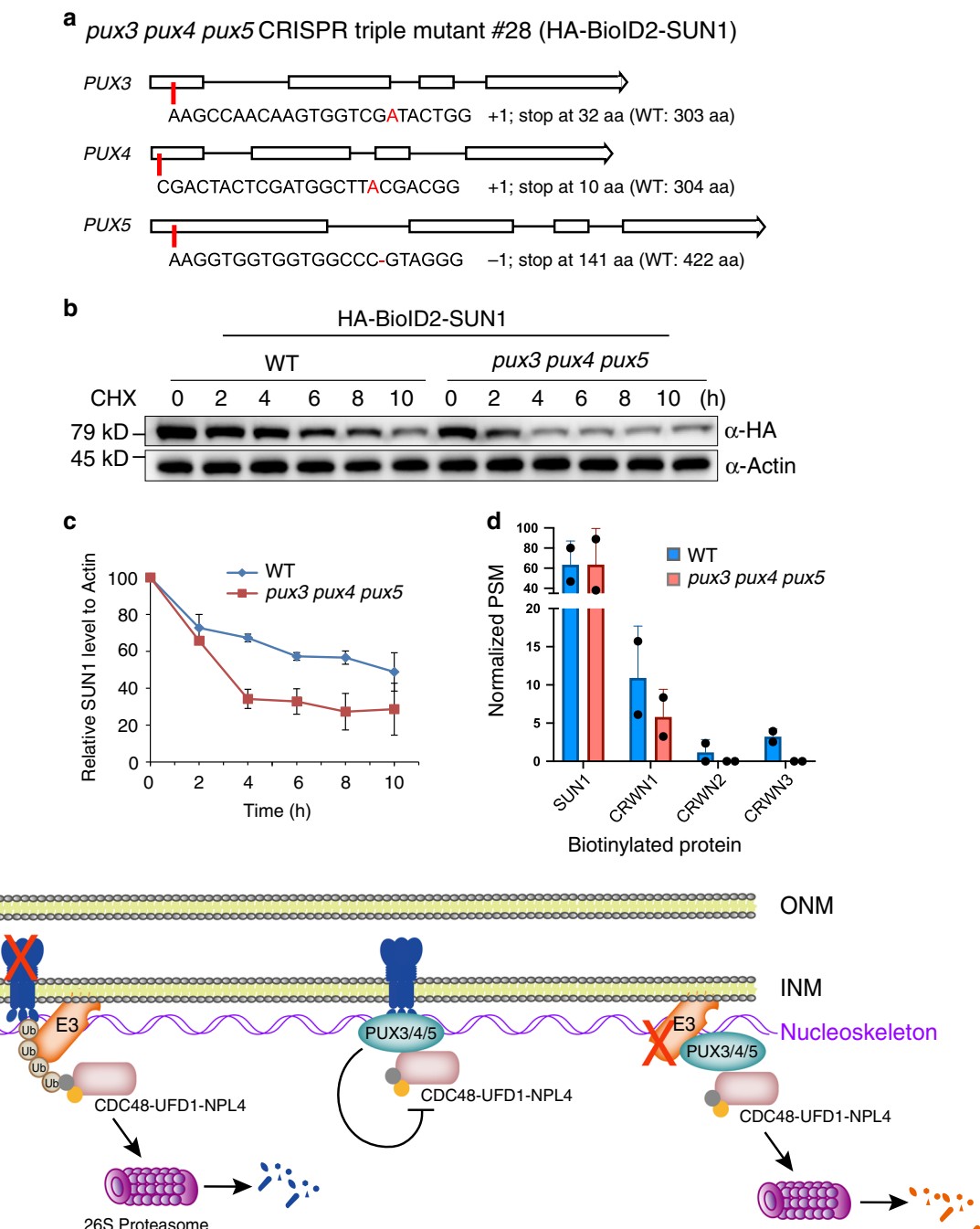

**Fig. 4 PUX3, PUX4, and PUX5 negatively regulate the degradation of SUN1. a** The *pux3, pux4,* and *pux5* mutations generated by CRISPR/Cas9 in the *35S: HA-BioID2-SUN1* background. **b** In vivo degradation assay for HA-BioID2-SUN1 in WT and the *pux3 pux4 pux5* triple mutant background. **c** Time-course densitometric analysis of the HA-BioID2-SUN1 protein level in the presence of CHX using actin as the control. Error bars represent SD from two independent biological replicates. **d** Peptide intensity values of nucleoskeleton proteins measured by PL-FLQMS using *35S: HA-BioID2-SUN1* plants in WT and *pux3 pux4 pux5* background. Error bars represent SD from two independent biological replicates. **e** Models for the function of PUX3/4/5 in plant INM protein degradation. INM proteins are ubiquitinated by unknown E3 ubiquitin ligases and subsequently recruit the CDC48 complex for proteasome degradation (left). PUX3/4/5 play a negative role in regulating the INM protein degradation in plants. One hypothesis is that PUX3/4/5 may protect INM proteins from non-specific degradation by directly interfering with the ATPase activity of CDC48 at the INM (middle). Alternatively, PUX3/4/5 may recruit the CDC48 complex to the INM to degrade the INM-localized E3 ligases that initiate INM protein degradation in plants (right). PUX3/4/5 are anchored to the INM through interacting with the nucleoskeleton.

membrane organelles in eukaryotes and have been under extensive investigation. In addition to the conserved and well-characterized ERAD pathway[36–38], chloroplast-associated protein degradation system (CHLORAD)[39] and lipid droplet-associated degradation (LDAD) pathway[21,22] were recently

discovered in plants. Here we report the identification of critical components for a ubiquitin/proteasome-dependent INM protein degradation pathway in plants using proximity-labeling proteomics. This pathway raises an important parallel with ERAD, CHLORAD, and LDAD pathways, all of which exploit the

CDC48 complex. It supports a converging theme that the ATP-driven chaperon plays a central role in mediating membrane-associated protein degradation at different organelles. Consistently, the CDC48 protein is wildly distributed throughout the cell, and the functionally characterized CDC48 ortholog in *Arabidopsis* (CDC48A) is essential for plant survival[40].

Although CDC48 is the common executor for the membrane-associated protein destruction, the substrate specificity and ATPase activity of CDC48 is subject to organelle-specific regulations. In particular, the functionally diversified PUX proteins appear to play an important role in this process. PUX10 has been reported to localize to lipid droplets and bridge CDC48 with membrane substrates in lipid droplets[32,33]. In contrast, PUX7/8/9/13 were shown to associate with autophagosomes and recruit inactive CDC48 for degradation[34]. Furthermore, PUX1 has been demonstrated to disrupt the CDC48 activity by directly promoting dissociation of the CDC48 hexamer[23,24]. Here, we found that another PUX subgroup (PUX3/4/5) is uniquely recruited to the INM through interacting with the nucleoskeleton underneath the INM. There, they negatively regulate the INM protein degradation by physically interacting with CDC48. We propose two possible models to explain the role of PUX3/4/5 during this process, and the two models are not mutually exclusive.

First, PUX3/4/5 may directly interfere with the CDC48 protein activity, like PUX1 (Fig. 4e). Because CDC48 was found highly concentrated in the nucleus and the nuclear periphery in *Arabidopsis*[40], PUX3/4/5 may function to inhibit the CDC48 activity at the INM and prevent non-specific degradation of INM proteins until they are ubiquitinated and ready for turnover. Second, PUX3/4/5 may mediate the degradation of putative E3 ubiquitin ligases that initiate the INM protein degradation in plants (Fig. 4e). In yeast, INM-associated E3 ligase Asi1 is critical for the ubiquitination of INM substrates for INMAD. The turnover of Asi1 itself depends on yeast UBX protein UBX1 and the CDC48 complex[35]. Because the PUX3/4/5 subclade bears homology with yeast UBX1 (Supplementary Fig. 3b), PUX3/4/5 may promote the stability of INM proteins through recruiting the CDC48 complex for degradation of INM-associated E3 ligases in plants. However, whether plant INM protein degradation is initiated by INM-associated E3 ubiquitin ligases, and if so, what the E3 ligases are, remain to be determined.

Lastly, the association of PUX3/4/5 with CRWN4 and KAKU4 suggests a nucleoskeleton-embedded regulatory mechanism for INM protein degradation in plants. The connection of the nucleoskeleton with the INM protein degradation is intriguing because it extends the functional importance of the nucleoskeleton to integral INM proteins, making it not only a scaffold for binding and retention of INM proteins[41] but also a platform that hosts molecules to regulate INM protein stability.

## Methods

**Plant growth conditions**. *Arabidopsis* (*Arabidopsis thaliana*) seeds were surface sterilized with 70% ethanol for 1 min and 20% (v/v) bleach for 8 min and rinsed with sterile water for five times before planted on the ½ Murashige and Skoog (½ MS) medium supplied with 1% agar. After 2 days of stratification at 4 °C, plants were moved to the plant growth room or chamber and grown under a photoperiod of 16 h-light and 8 h-dark at 22 °C. For proximity labeling and mass spectrometry, 10-day-old seedlings were used. *Nicotiana benthamiana* plants used for transient protein expression were grown under the same conditions except that the seeds were planted directly into the soil.

**Vector construction for proximity labeling**. Cloning was performed using a multisite gateway system as reported before[42,43]. Briefly, *HA-BioID2* and *PUX5* were cloned into the multisite gateway entry vector pBSDONR p1-p4 using BP reaction (BP clonase™ II, Thermo Fisher, Cat#11789020). *BioID2-HA*, *WIT1*, *RHD3*, and *NEMP_A* were cloned into pBSDONR p4r-p2. BioID2 fusions were generated by fusing constructs in pBSDONRp1-p4 (N-terminus) and pBSDONRp4r-p2 (C-terminus) into the destination vector pEarlyGate100 by LR

reaction (LR clonase™ II plus, Thermo Fisher, Cat#12538200). All primers used in this study were listed in Supplementary Data 1.

**Plant materials**. *Arabidopsis thaliana* ecotype Columbia (Col-0) was used as the wild-type (WT). Transgenic *Arabidopsis* lines used for proximity labeling (*35S: HA-BioID2-WIT1, 35S: HA-BioID2-NEMP_A, 35S: HA-BioID2-RHD3*, and *35S: PUX5-BioID2-HA*) were generated by floral dip transformation of WT plants with Agrobacterium strain GV3101 carrying corresponding constructs. T1 transgenic plants were selected by Basta resistance, and the protein expression level of individual lines was examined by immunoblotting using streptavidin-HRP (Abcam, Cat#7403, dilution 1:10,000) and anti-HA antibody (3F10, Roche, Cat#11867431001, dilution 1:5000) using T2 seedlings treated with 50 μM biotin. The *35S: HA-BioID2-SUN1* transgenic line was described before[16].

**Proximity labeling and free biotin depletion**. For proximity labeling, 10-day-old transgenic seedlings expressing protein tagged with BioID2 were submerged in 50 μM biotin solution for 16 h at room temperature before being washed three times with ice-cold water and harvested. Total protein was extracted by grinding the sample with extraction buffer [50 mM Tris-HCl (pH 7.5), 150 mM NaCl, 0.5% NP40 (v/v), 0.5% Triton X-100 (v/v), 0.5% sodium deoxycholate (w/v), 40 μM MG132, 1× Protease Inhibitor Cocktail]. The supernatant was collected after samples were centrifuged at 13,000 rpm at 4 °C for 10 min.

**Removal of free biotin by desalting chromatography**. For desalting chromatography, the AKTA protein purification system (GE Healthcare) with 3 × 5 mL HiTrap desalting columns (GE Healthcare, Cat#GE29-0486-84) was utilized. About 2 mL of total proteins for each sample were loaded into the HiTrap column after equilibration with desalting buffer (50 mM Tris-HCl pH 7.5, 0.05% Triton X-100) using 5 mL syringe with a 2 mL sample loop. The flow rate used for all experiments was 0.7 mL/min with 0.3 Mpa as the pressure limit. The salt elute containing free biotin with high conductivity were abandoned while the total protein elute with UV280 absorbance peak was collected for affinity purification (AP).

**Affinity purification by streptavidin-coated beads**. About 3 mL desalted protein elute was collected for each sample and mixed with 50 μL streptavidin-coated magnetic beads (Dynabeads™ Myone™ Streptavidin C1, Thermo Fisher, Cat#65002) for AP. After incubation with rotation at 4 °C overnight, the beads were separated on a magnet rack and washed with extraction buffer for five times. Proteins on beads were then eluted by boiling at 98 °C for 20 min in elution buffer [50 mM Tris-HCl (pH 7.5), 150 mM NaCl, 1% SDS, 50 μM biotin] before separated by SDS-PAGE. Each lane of SDS-PAGE gel was cut into four individual pieces and digested with trypsin in 50 mM ammonium bicarbonate at 37 °C overnight, extracted twice with 1% formic acid in 50% acetonitrile aqueous solution, and dried by SpeedVac before LC-MS/MS analysis was performed.

**LC-MS/MS**. For LC-MS/MS analysis, tryptic peptides were separated by a 135 min gradient elution at a flow rate 0.300 μL/min with the DIONEX ultimate 3000 integrated nano-HPLC system (Thermo Fisher) which is directly interfaced with the Thermo LTQ-Orbitrap mass spectrometer. The analytical column was Acclaim PopMap™ 100 C18 capillary column (75 μm ID, 150 mm length, Thermo Fisher Scientific, Cat#164568) packed with C-18 resin (300 Å, 5 μm, Varian). Mobile phase A consisted of 0.1% formic acid, and mobile phase B consisted of 100% acetonitrile and 0.1% formic acid. The LTQ-Orbitrap mass spectrometer was operated in the data-dependent acquisition mode using the Xcalibur 4.1 software. There is a single full-scan mass spectrum in the Orbitrap (400–1800 $m/z$, 30,000 resolution) followed by 20 data-dependent MS/MS scans in the ion trap at 35% normalized collision energy. MS/MS spectra from each LC-MS/MS run were searched against the TAIR10 database using Proteome Discoverer (Version 2.2; Thermo Fisher) with the following criteria: full tryptic specificity was required; two missed cleavages were allowed; carbamidomethylation was set as fixed modification; oxidation (M) were set as variable modifications; precursor ion mass tolerance was 20 ppm for all MS acquired in the Orbitrap mass analyzer; and fragment ion mass tolerance was 0.02 Da for all MS spectra. High confidence score filter (FDR < 1%) was used to select the "hit" peptides and their corresponding MS/MS spectra were manually inspected.

**MS data analysis**. For proximity biotinylation data analysis, protein enrichment areas (LFQ values) were integrated and used as the input for sample normalization by DEP package (version 1.8.0) in RStudio (version 1.1.463). Specifically, the output file from Proteome Discoverer (Version 2.2) was imported into DEP using the LFQ intensities as main category. The data matrix was filtered to remove proteins with missing values using less stringent filtering (thr = 1) and then was background corrected and normalized by variance stabilizing transformation. The remaining missing values in the dataset were further imputed (fun = "MinProb", $q = 0.01$) for missing not at random (MNAR), which indicate that proteins are below the detection limit in specific samples (e.g., in the control samples). Then differential enrichment analysis which is based on linear models and empirical Bayes statistics was performed with manually specified contrasts. Significant

proteins are determined by defined cutoffs using fold-change and *p*-vaule compared to controls.

**Yeast two-hybrid assays**. The CDS of *PUX4/5, CDC48A/B/C*, and *UFD1B/1C* were fused to the GAL4-activation domain or GAL4-binding domain and cloned into the pGBKT7 or pGADT7 vectors by seamless ligation with ClonExpress II One Step Cloning Kit (Vazyme, Cat#C112). All fusion constructs were confirmed by sequencing. These constructs were transformed into *Saccharomyces cerevisiae* strain Y187 and Y2HGold (Yeastmaker Yeast Transformation System 2, Clontech), and transformants were selected using colony PCR. Y187 and Y2HGold were mated in 4 mL 2× YPDA medium at 30 °C for 18–22 h. The resulting culture containing diploid yeasts was diluted and dropped on DDO medium (SD-Leu/-Trp) and QDO medium (SD-Leu/-Trp/-His/-Ade) and incubated at 30 °C for 3 days before photos were taken. Autoactivation assays have been performed for each bait and prey construct with corresponding empty vector to exclude potential false positives.

**In vivo protein degradation assay**. Ten-day-old seedlings were treated by 100 μM CHX with or without 50 μM MG132 or 1 μM Concanamycin A for the indicated time. About 20 seedlings were sampled per treatment per time point. The samples were dried with paper towels and weighed to keep the biomass the same for each sample. Samples were then frozen in liquid nitrogen and ground to fine powder. Protein was extracted by adding 200 μL extraction buffer [50 mM Tris-HCl (pH7.5), 150 mM NaCl, 0.5% NP40 (v/v), 0.5% Triton X-100 (v/v), 0.5% sodium deoxycholate (w/v), 40 μM MG132, 1× Protease Inhibitor Cocktail] to the fine powder and vortex mixing. The supernatant was collected after centrifuged at 13,000 rpm at 4 °C for 10 min. Protein samples were adjusted to a final concentration of 300 mg/mL, separated by SDS-PAGE, and subject to immunoblotting using anti-HA (3F10, Roche, Cat#11867431001, dilution 1: 5000) and anti-Actin antibody (Abiocode, Cat#R3772-1P, dilution 1: 3000).

**CRISPR/Cas9-mediated knockout of *PUX3/4/5***. The single guide RNA (sgRNA) sequences that target *PUX3, PUX4*, and *PUX5* were designed using the webserver CRISPOR (http://crispor.tefor.net/). Using pCBC-DT1T2 as the template, the sgRNA (*PUX4*)-U6-26t-U6-29p-sgRNA (*PUX5*) cassette was amplified by PCR and inserted into pHEE401 by GoldenGate assembly to obtain pHEE401-*PUX4/PUX5*. The sgRNA (*PUX3*) was cloned into pHEE401, which was then digested with BsaI to obtain the U6-26p-sgRNA (PUX3)-U6-29t fragment. This fragment was then introduced into the pHEE401-*PUX4/PUX5* by seamless cloning to obtain pHEE401-*PUX3/PUX4/PUX5* construct. Another plasmid with a different set of sgRNAs for *PUX3/PUX4/PUX5* was constructed in a similar manner. The two pHEE401-*PUX3/PUX4/PUX5* plasmids were transformed into the homozygous T3 lines of *35S: HA-BioID2-SUN1*. Mutants that contain different combinations of *PUX3/PUX4/PUX5* mutations were identified in the T1 generation by amplifying and sequencing of the genomic fragments of *PUX3/PUX4/PUX5* genes. Homozygous mutations were obtained and confirmed by sequencing in the T2 generation.

**Co-immunoprecipitation and BiFC**. Co-IP and BiFC experiments were performed using transient protein expression in *N. benthamiana* leaves infiltrated with Agrobacterium carrying corresponding constructs as described previously[44]. For co-IP, a 15 μL volume of HA antibody (3F10, Roche) was added to 1 mL of total protein extract and incubated at 4 °C for 4 h with gentle shaking before 20 μL of pre-rinsed protein G agarose beads (Millipore) was added. The mixture was incubated for another 3 h at 4 °C. The beads were washed five times with PBS buffer, separated by SDS-PAGE, and subjected to immunoblotting using anti-GFP (Living Colors, Takara, Cat#632375, dilution 1:3000), anti-HA antibody (3F10, Roche, Cat#11867431001, dilution 1:5000), or anti-ubiquitin antibody (D9D5, Cell Signaling Technology, Cat#8081S, dilution 1:1000). For BiFC, leaves were collected for microscopic imaging 48 h after Agrobacterium infiltration. Confocal laser scanning microscopy was performed using a TCS SP8 STED confocal microscope (Leica). sYFP fluorescence was excited by the 514 nm Argon laser and detected using a custom 522 nm to 545 nm band-pass emission filter, whereas mCherry fluorescence was excited by the 561 nm laser and detected using a custom 595–620 nm band-pass emission filter.

**Immunogold labeling and electron microscopy**. Roots of homozygous 35S: *HA-BioID2-SUN1* T3 transgenic plants were excised and immersed in 20% (w/v) BSA and frozen immediately in a high-pressure freezer (HPM100, Leica). Samples were put into tubes containing 0.2% uranyl acetate in acetone. Freeze substitution and low-temperature embedding were carried out using Leica EM AFS2 as follows: −90 °C for 72 h, 2 °C/h increase for 15 h, −60 °C for 8 h, 2 °C/h increase for 15 h, −30 °C for 8 h, and 2 °C per hour increase for 5 h to −20 °C. Samples were then washed 3 times with 100% ethanol for 3 h, infiltrated stepwise over 3 days at −20 °C in LR Gold resin and embedded in capsules. The polymerization was performed at −20 °C for 24 h and room temperature for 3 days using UV light. Ultrathin sections were made using an ultramicrotome (Leica EM UC6). Samples

were treated with anti-HA primary antibody (3F10, Roche, Cat#11867431001, dilution 1:10) and 12 nm gold-conjugated goat anti-rat IgG (1:20, Jackson ImmunoResearch) on the grids and post-stained in uranyl acetate for 30 min and in lead citrate for 3 min. Grids were imaged with a transmission electron microscope (H-7650B, Hitachi).

**Reporting summary**. Further information on research design is available in the Nature Research Reporting Summary linked to this article.

## Data availability

All data generated in this study has been made available either in the Source Data file, via respective repository entry, or Supplementary Information files and are available from the corresponding author on reasonable request. Relevant mass spectrometry (MS) proteomics data have been deposited to the ProteomeXchange Consortium via the PRIDE partner repository (Identifier: PXD015920). The MS raw data for BioID2-SUN1, BioID2-NEAP1, Nup93a-BioID2, Nup82-BioID2, BioID2-WIP1, and Mock control were retrieved from PXD015919[16]. The MS datasets and figures associated with each dataset are listed in Supplementary Data 2. Differential analysis and normalized MS data for BioID2-SUN1, BioID2-WIT1, and PUX5-BioID2 proximity-labeling proteomic experiments were provided in Supplementary Data 3.

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

## Acknowledgements

We thank Dr. Sheila McCormick for critical reading of the manuscript and Dr. Qi-Jun Chen for providing the pCBC-DT1T2 and pHEE401 vector. This research was supported by grants from Innovative Genomics Institute at the University of California Berkeley and Tsinghua-Peking Joint Center for Life Sciences at Tsinghua University.

## Author contributions

A.H. and Y.G. designed the research. A.H., Y.T., X.S., M.J., J.Z., X.Y., and H.C. performed the experiments. A.H. and Y.G. analyzed the data and wrote the paper.

## Competing interests

The authors declare no competing interests.
