## [Peer Review File · Nature Communications]

Reviewers' comments:

Reviewer #1 (Remarks to the Author):

The manuscript by Huang et al. utilizes BioID2 based proximity labeling to reveal proteins that function to regulate inner nuclear membrane-associated protein degradation in plants. They begin by probing interactors of the SUN1 protein, which is well characterized nuclear envelope protein. They go on to identify and characterize the INMAD complex and identify factors that negatively regulate INMAD (PUX) using a series of biochemical and/or genetic experiments in yeast and in plants. Together the authors identify and nicely characterize components of INMAD in plants.

- The authors should better contextualize their study with regards to previous proximity labeling papers. For example, in the introduction they state: "By combining the power of proximity labeling technology¹⁰, label-free quantitative mass spectrometry, and ratiometric analysis, we developed proximity-labeling proteomics in plants.". However, proximity labeling has already been developed for plants including BioID (Lin et al <https://doi.org/10.3389/fpls.2017.00749>; Khan et al., <https://doi.org/10.1038/s41598-018-27500-3>; Conlan et al., <https://doi.org/10.3389/fpls.2018.01882>). Furthermore, improved proximity labeling approaches using TurboID, which outperforms BioID and BioID2, have also been developed (Zhang et al., <https://doi.org/10.1038/s41467-019-11202-z>; Mair et al., DOI: 10.7554/eLife.47864).

- Thus, the authors may want to discuss the possibility that the proximal proteins they identified using BioID2 may not be an exhaustive list of all possible interactors (due to low biotinylation activity of BioID2). I want to emphasize that this is not a major critique of the manuscript, which nicely utilizes BioID2 PL to reveal proteins that negatively regulate INMAD in plants.

- First paragraph of the results states "We transformed the 35S: HA-BioID2-SUN1 construct into Arabidopsis and selected a transgenic line with a relatively low expression level.". No data is shown documenting this. Also, is it mRNA expression level or protein level?

- Figure S1a. Please indicate the temperature used for the 16 h biotin labeling. A major criticism of BioID and BioID2 is weak labeling at temperatures lower than 37 C.

- Sufficient information to assess the proteomics data is lacking.

• Detailed information on HPLC separation gradient and mass spectrometer acquisition settings is missing and are standard requirements for proteomics experiments (this is lacking in the manuscript under review elsewhere file as well).

• What settings and software were used to search the mass spectra? A supplemental file listing all proteins detected, their identification score, and abundance of the proteins in each individual replicate would also help assess the data quality; rather than just the summary tables such as Fig S2.

• The ProteomeXchange dataset is not accessible to reviewers.

• Fig 2a suggests that BioID2-SUN1 is expressed at a lower level and the BioID2 control. This has been observed in other proximity labeling reports so its not unexpected. However, more information in the methods would clarify how this may or may not have impacted the determination of enriched interactors. Were equal volumes resulting from the in gel digest used for LCMS analysis or equal peptide? Were the protein abundance values normalized prior to calculating FC and enrichment?

• How were statistical tests to determine enriched interactors performed?

- Showing empty AD/BD alone with each prey/bait would strengthen figure 1d.

- Fig.2: In general Actin is a poor loading control for Western Blots (Proteomics: 2017 Oct;17(20). doi: 10.1002/pmic.201600189.) as it is easy to saturate etc. However, Fig 2 c,d,e are convincing

that SUN1 is degraded in a proteasome dependent manner.

- "highest" should be removed from the following sentence on page 6: "...the PUX3/4/5 subgroup bears the highest homology to the yeast UBX protein Ubx1 (Supplementary Fig. 3b),...". There is no degree of homology there is either a shared evolutionary ancestry or not. Highest homology should also be corrected in the discussion.

- Perhaps I'm missing something but did the treatments work in Fig S5c? There is no apparent decrease in SUN1 levels following 10 hours of CHX (in the - MG132 lane), which is in contrast to Fig 4b and Fig S5b. This calls into question this conclusion stated on pg8: "Nevertheless, MG132 treatment abolished the SUN1 degradation in the pux3 pux4 pux5 triple mutant, suggesting that the regulation by PUX3/4/5 is upstream of the proteasome (Supplementary Fig. 5c).".

Reviewer #2 (Remarks to the Author):

This work by Gu et al. describes protein degradation at the inner nuclear membrane. The inner nuclear membrane degradation (INMAD) pathway, part of the ubiquitin proteasome system, functions in fungi and animals, but currently is unclear whether it acts in plants.

The authors have determined that AtSUN1 is a nonstable protein, and likely subject to INMAD regulation. Using proximity labeling and mass spectrometry-based proteomics, they determined that AtSUN1 interacts with Cdc48/p97, which is thought to relay ubiquitinated substrates to the proteasome. Importantly, the authors identified a class of CDC48/p97 binding factors called PUXs that are associated with AtSUN1. Most convincing is that PUXs negatively regulates AtSUN1 stability. These observations are novel, and to my knowledge, represent the first example of a plant SUN protein that is degraded by the INMAD pathway. Their findings therefore extend our knowledge of protein quality control at the nuclear membrane to plants, and is of great interest to the broad field of nuclear envelope biology.

The experiments described in this manuscript are well designed, described in detail, and the data appear to be in high quality. My major concern of this manuscript, as in its current form, is the authors' over interpretation of their data. INMAD is a protein degradation pathway that works specifically at the INM. Definition of the INMAD pathway relies on the E2 ubiquitin conjugating enzymes and the E3 ubiquitin ligases, the latter of which determine substrate specificity. While it is convincing that AtSUN 1 is a nonstable protein and is presumably degraded by the proteasome, there is no direct evidence that AtSUN1 is an INMAD substrate. For example, (1) whether AtSUN1 is ubiquitinated is unknown, (2) If AtSUN1 is ubiquitinated, what are the responsible E2 and E3 enzymes? and (3) whether CDC48/p97 directly regulates AtSUN1 stability is unclear. With none of these issues addressed, it is overstating that there is an INMAD pathway in plants.

Response to Reviewers' comments (Remarks to the Author / Authors' responses):

Reviewer #1:

The manuscript by Huang et al. utilizes BioID2 based proximity labeling to reveal proteins that function to regulate inner nuclear membrane-associated protein degradation in plants. They begin by probing interactors of the SUN1 protein, which is well characterized nuclear envelope protein. They go onto identify and characterize the INMAD complex and identify factors that negatively regulate INMAD (PUX) using a series of biochemical and/or genetic experiments in yeast and in planta. Together the authors identify and nicely characterize components of INMAD in plants.

We thank the reviewer for appreciating the quality and significance of our work.

- The authors should better contextualize their study with regards to previous proximity labeling papers. For example, in the introduction they state: "By combining the power of proximity labeling technology¹⁰, label-free quantitative mass spectrometry, and ratiometric analysis, we developed proximity-labeling proteomics in plants.". However, proximity labeling has already been developed for plants including BioID (Lin et al <https://doi.org/10.3389/fpls.2017.00749>; Khan et al., <https://doi.org/10.1038/s41598-018-27500-3>; Conlan et al., <https://doi.org/10.3389/fpls.2018.01882>). Furthermore, improved proximity labeling approaches using TurboID, which outperforms BioID and BioID2, have also been developed (Zhang et al., <https://doi.org/10.1038/s41467-019-11202-z>; Mair et al., DOI: 10.7554/eLife.47864).

We added the following sentence to the introduction "Other proximity-labeling based proteomic approaches have been recently applied to plant research and have been shown to be invaluable in profiling functional components of various protein complexes", and these works are now appropriately cited. We also modified the text to specify that we independently developed our own proximity-labeling proteomic approach in plants that we named PL-LFQMS-RA.

- Thus, the authors may want to discuss the possibility that the proximal proteins they identified using BioID2 may not be an exhaustive list of all possible interactors (due to low biotinylation activity of BioID2). I want to emphasize that this is not a major critique of the manuscript, which nicely utilizes BioID2 PL to reveal proteins that negatively regulate INMAD in plants.

We allowed the BioID2 to label for 16 hours, which is close to reaching the plateau for BioID2-mediated biotinylation in our hand. However, we agree with the reviewer that TurboID has been shown to outperform BioID2 by Zhang et al. in plants and can label substrates much faster than BioID2. TurboID-mediated biotinylation also appears to be more extensive than BioID2 when the labeling is saturated. We added the following sentences to the text "An improved biotin ligase with higher labeling efficiency termed TurboID has been recently developed (Baron et al., 2018, Mair et al., 2019, Zhang et al., 2019). Using TurboID in PL-LFQMS-RA may further improve its efficiency and result in a more comprehensive profiling."

- First paragraph of the results states “We transformed the 35S: HA-BioID2-SUN1 construct into Arabidopsis and selected a transgenic line with a relatively low expression level.”. No data is shown documenting this. Also, is it mRNA expression level or protein level?

The original western blot for the screen of expression levels of different 35S: HA-BioID2-SUN1 lines has now been added to Figure S1b. We have also specified in the text that the low expression is based on the protein level.

- Figure S1a. Please indicate the temperature used for the 16 h biotin labeling. A major criticism of BioID and BioID2 is weak labeling at temperatures lower than 37 C.

We used room temperature instead of 37°C for biotin labeling to avoid potential changes in interactome induced by high-temperature stress. This information has been added to Figure S1a.

- Sufficient information to assess the proteomics data is lacking:

- Detailed information on HPLC separation gradient and mass spectrometer acquisition settings is missing and are standard requirements for proteomics experiments (this is lacking in the manuscript under review elsewhere file as well).

Please see the FPLC information in the newly added section “Removal of free biotin by desalting chromatography” in the Methods. Briefly, ~2 mL of total proteins for each sample were loaded into the HiTrap column after the equilibration with desalting buffer (50 mM Tris-HCl pH 7.5, 0.05% Triton X-100) using 5 mL syringe with a 2 mL sample loop. The flow rate used for all experiments was 0.7 mL/min with 0.3 Mpa as the pressure limit. The salt elute containing free biotin with high conductivity were abandoned while the total protein elute (~3 mL) with UV280 absorbance peak was collected for affinity purification. Please see the MS acquisition settings below.

- What settings and software were used to search the mass spectra?

For LC-MS/MS analysis, tryptic peptides were separated by a 60 min gradient elution at a flow rate 0.25 µL/min with the DIONEX ultimate 3000 integrated nano-HPLC system (Thermo Fisher) which is directly interfaced with the Thermo LTQ-Orbitrap mass spectrometer. The analytical column was a home-made fused silica capillary column (75 µm ID, 150 mm length; Upchurch Scientific) packed with C-18 resin (300 Å, 5 µm; Varian). Mobile phase A consisted of 0.1% formic acid, and mobile phase B consisted of 100% acetonitrile and 0.1% formic acid. The LTQ-Orbitrap mass spectrometer was operated in the data-dependent acquisition mode using the Xcalibur 2.0.7 software and there is a single full-scan mass spectrum in the Orbitrap (400-1800 m/z, 30,000 resolution) followed by 20 data-dependent MS/MS scans in the ion trap at 35% normalized collision energy.

MS/MS spectra from each LC-MS/MS run were searched against the TAIR10 database using Proteome Discoverer (Version 2.2; Thermo Fisher) with the following criteria: full tryptic specificity was required; two missed cleavages were allowed; carbamidomethylation was set as fixed modification; oxidation (M) were set as variable modifications; precursor ion mass tolerance was 20 ppm for all MS acquired in the Orbitrap mass analyzer; and fragment ion mass tolerance was 0.02 Da for all MS spectra. High confidence score filter (FDR < 1%) was used to select the “hit” peptides and their corresponding MS/MS spectra were manually inspected.

The above information has been added to the new “LC-MS/MS” section in the Methods.

A supplemental file listing all proteins detected, their identification score, and abundance of the proteins in each individual replicate would also help assess the data quality; rather than just the summary tables such as Fig S2.

We added a Supplementary Table 3 to include the requested raw data information for BioIDs-SUN1, BioID2-WIT1, and PUX5-BioID2 MS datasets reported in this manuscript.

The ProteomeXchange dataset is not accessible to reviewers.

PXD015920 is now publicly available for download through ProteomeXchange.

• Fig 2a suggests that BioID2-SUN1 is expressed at a lower level and (than) the BioID2 control. This has been observed in other proximity labeling reports so its not unexpected. However, more information in the methods would clarify how this may or may not have impacted the determination of enriched interactors. Were equal volumes resulting from the in gel digest used for LCMS analysis or equal peptide? Were the protein abundance values normalized prior to calculating FC and enrichment?

Yes, we used equal volume samples that come from in-gel digestion for LC-MS/MS analysis and normalized the data before enrichment analysis. For normalization, protein enrichment areas (LFQ values) were integrated and used as the input for normalization among samples using variance stabilizing transformation (DEP package in R). The normalized LFQ values were then used as the input for subsequent differential enrichment analysis. Please see more details in the newly added “MS data analysis” section in the Methods.

Our controls and analysis method appear to be very efficient. Within the 15 and 4 high-confidence candidates identified for BioID2-SUN1 and BioID2-WIT1, respectively, we found almost all previously reported interactors for SUN1 (CRWN1, CRWN2, CRWN3, CRWN4, and KAKU4) and WIT1 (WIP1, WIP3, RanGAP1, and RanGAP4) (Fig.1b). For SUN1, we also identified another two bona fide membrane NE proteins that we reported in the accompanying paper (PNET7 and PNET10). Therefore, the PL-LFQMS-RA method-determined candidate lists are of high quality.

• How were statistical tests to determine enriched interactors performed?

Statistical inferences were obtained using differential enrichment analysis based on linear regression models and empirical Bayes statistics (DEP package in R). Significantly enriched proteins are determined by defined cutoffs (described in figures and figure legends) using fold-change and *p*-value compared to controls.

- Showing empty AD/BD alone with each prey/bait would strengthen figure 1d.

We tested autoactivation for each prey and bait construct with empty vectors before they were used for Y2H assays to exclude potential false positives. The original results of autoactivation are attached below, and relevant parts have now been incorporated in Fig. 1d.

- Fig.2: In general Actin is a poor loading control for Western Blots (Proteomics: 2017 Oct;17(20). doi: 10.1002/pmic.201600189.) as it is easy to saturate etc. However, Fig 2 c,d,e are convincing that SUN1 is degraded in a proteasome dependent manner.

Thank you for the useful suggestion. As the reviewer has perceived in this case, even the Actin is not a perfect loading control, the SUN1 degradation is clearly proteasome-dependent.

- “highest” should be removed from the following sentence on page 6: “...the PUX3/4/5 subgroup bears the highest homology to the yeast UBX protein Ubx1 (Supplementary Fig. 3b),...”. There is no degree of homology there is either a shared evolutionary ancestry or not. Highest homology should also be corrected in the discussion.

We have removed “the highest” in the text. Thank you.

- Perhaps I’m missing something but did the treatments work in Fig S5c? There is no apparent decrease in SUN1 levels following 10 hours of CHX (in the – MG132 lane), which is in contrast to Fig 4b and Fig S5b. This calls into question this conclusion stated on pg8: “Nevertheless, MG132 treatment abolished the SUN1 degradation in the *pux3 pux4 pux5* triple mutant, suggesting that the regulation by PUX3/4/5 is upstream of the proteasome (Supplementary Fig. 5c).”

Thank you very much for pointing out the problem with Fig. S5c, which may be due to a failed CHX treatment. We have now repeated this experiment and included the new results in Fig. S5c. MG132 greatly compromised SUN1 degradation in the *pux3 pux4 pux5* plants, similar to

what was observed in WT, supporting our conclusion that PUX3/4/5 function upstream of the proteasome.

Reviewer #2 (Remarks to the Author):

This work by Gu et al. describes protein degradation at the inner nuclear membrane. The inner nuclear membrane degradation (INMAD) pathway, part of the ubiquitin proteasome system, functions in fungi and animals, but currently is unclear whether it acts in plants.

The authors have determined that AtSUN1 is a nonstable protein, and likely subject to INMAD regulation. Using proximity labeling and mass spectrometry-based proteomics, they determined that AtSUN1 interacts with Cdc48/p97, which is thought to relay ubiquitinated substrates to the proteasome. Importantly, the authors identified a class of CDC48/p97 binding factors called PUXs that are associated with AtSUN1. Most convincing is that PUXs negatively regulates AtSUN1 stability.

These observations are novel, and to my knowledge, represent the first example of a plant SUN protein that is degraded by the INMAD pathway. Their findings therefore extend our knowledge of protein quality control at the nuclear membrane to plants, and is of great interest to the broad field of nuclear envelope biology.

The experiments described in this manuscript are well designed, described in detail, and the data appear to be in high quality.

We thank the reviewer for appreciating the quality, novelty, and significance of our work.

My major concern of this manuscript, as in its current form, is the authors' over interpretation of their data. INMAD is a protein degradation pathway that works specifically at the INM. Definition of the INMAD pathway relies on the E2 ubiquitin conjugating enzymes and the E3 ubiquitin ligases, the latter of which determine substrate specificity. While it is convincing that AtSUN 1 is a nonstable protein and is presumably degraded by the proteasome, there is no direct evidence that AtSUN1 is an INMAD substrate. For example, (1) whether AtSUN1 is ubiquitinated is unknown, (2) If AtSUN1 is ubiquitinated, what are the responsible E2 and E3 enzymes? and (3) whether CDC48/p97 directly regulates AtSUN1 stability is unclear. With none of these issues addressed, it is overstating that there is an INMAD pathway in plants.

(1) SUN1 is indeed ubiquitinated, especially in the presence of MG132. The data was shown in Fig. 2b; (2) We agree with the reviewer that our manuscript did not investigate potential E3 ligases involved, which is our next research focus but out of the scope of this manuscript. We have revised the title, abstract, and main text to avoid potential overstatement on discovery of "the INMAD pathway" and used "an INM protein degradation pathway" instead. However, the CDC48-mediated proteasome-dependent INM protein degradation pathway represents a possible and likely an important candidate for the plant INMAD pathway. It raises a critical parallel with the yeast INMAD pathway and also other recently identified organelle-specific and

membrane-associated protein degradation pathways in plants, including the chloroplast-associated degradation pathway (CHLORAD, Science 2019, 363:836) and the lipid droplet-associated degradation pathway (LDAD, Plant Cell 2018, 30:2116 and 30:2137).

We also want to emphasize that the association with the CDC48 complex and PUX3/4/5 is not restricted to SUN1 protein but applies to two other known INM proteins as well (Fig. 1e), suggesting the reported pathway may work for degradation of other INM proteins as well. Moreover, the INM targeting mechanism of PUX3/4/5 does not rely on specific INM proteins but uses the nucleoskeleton to approach INM proteins in general, which supports their potential function in regulating the stability of a broad spectrum of substrates at the INM.

To answer the third question: as we discussed in the manuscript, CDC48 is widely distributed throughout the cell and appears to play a central role in multiple membrane-associated degradation systems other than INMAD in plants, including ERAD, CHLORAD, and LDAD. Therefore, using CDC48 complex mutant to study the INMAD-specific effect may yield confounding results. More importantly, knocking out of the characterized functional Arabidopsis CDC48 ortholog (CDC48A) results in seedling lethality. Consistently, we mutagenized UFD1B and UFB1C (essential CDC48 complex cofactors) in Arabidopsis using CRISPR and found the *ufd1b ufd1c* double mutant is also lethal, likely due to compromised CDC48 complex as well. These results indicate that the CDC48 complex is functionally essential in plants, which precludes us from using traditional mutant analysis to demonstrate its specific effect on the INM protein degradation. However, the proteasome dependency of the discovered INM proteolysis pathway (Fig. 2c-e) is consistent with the fact that CDC48 works with the proteasome for membrane protein degradation, and MG132 treatment indeed trapped the CDC48 complex to its INM substrate (Fig. 2f), supporting its role in degrading INM proteins. More importantly, we found a group of CDC48-interacting PUX proteins (PUX3/4/5) at the INM and demonstrated their role in the INM protein degradation. These evidence all support a direct role of CDC48 in INM protein degradation.

REVIEWERS' COMMENTS:

Reviewer #1 (Remarks to the Author):

The authors have worked to nicely address my comments. I'm still not clear how "PL-LFQMS-RA" is any different from many previous proximity labeling reports that use label-free based quantification. This is a minor point as their use of proximity labeling to gain mechanistic insight into INMAD is the main feature of the paper (method novelty is an oversell). I am fine with the authors referring to their method as PL-LFQMS-RA.

Reviewer #2 (Remarks to the Author):

The authors have addressed my major concerns.

Reviewers' Comments / Authors' Responses

Reviewer #1 (Remarks to the Author):

The authors have worked to nicely address my comments. I'm still not clear how "PL-LFQMS-RA" is any different from many previous proximity labeling reports that use label-free based quantification. This is a minor point as their use of proximity labeling to gain mechanistic insight into INMAD is the main feature of the paper (method novelty is an oversell). I am fine with the authors referring to their method as PL-LFQMS-RA.

We have modified the introduction according to the reviewer's suggestion to avoid over-claiming about method novelty. It is worth to mention that we are pioneered in using ratiometric analysis for sample analysis in plant research, which helped to make substantial difference in achieving low false positive rate and high specificity in candidate identification.

Reviewer #2 (Remarks to the Author):

The authors have addressed my major concerns.

Thanks.